# Towards Text–Mask Consistency in Medical Image Segmentation

**Jie Gui**[1,2]**, Hang Tu**[1,2]**, Wen Sha**[1,2]***, Xiuquan Du**[1,3]*

[1]Key Laboratory of Intelligent Computing and Signal Processing, Ministry of Education, Anhui University, Hefei, China
[2]School of Artificial Intelligence, Anhui University, Hefei, China
[3]School of Computer Science and Technology, Anhui University, Hefei, China
`{w125111032,w125121009}@stu.ahu.edu.cn`
`{04069,dxqllp}@ahu.edu.cn`

## Abstract

Vision-language models for medical image segmentation often produce masks that conflict with the accompanying text, especially under multi-site/multi-lesion descriptions. We trace this failure to two factors: (i) highly templated and repetitive clinical language causes one-to-one hard contrastive learning to yield numerous false negatives, weakening cross-modal alignment; and (ii) predominantly vision-driven, one-way cross-attention lacks a language-dominant, spatially aware pathway, hindering effective injection of textual semantics into the spatial visual domain. To this end, we propose Consistency-enhanced Two-stage Segmentation (**C2Seg**). In the pretraining stage, Cluster-aware Contrastive Learning uses a frozen strong baseline to construct an intra-batch text similarity matrix as soft labels, thereby alleviating false negative conflicts and producing more discriminative visual representations. In the fusion stage, we introduce a Bidirectional Complementary Attention Module, where each modality dominates attention along its own path, fostering deep interaction and structural consistency between visual and textual representations. In order to enhance the expressive power of multimodal features, we further adopt KAN-based Attention Gating. Without updating the language encoder, our approach significantly improves text–mask consistency and segmentation accuracy on four public medical imaging datasets.

## 1 Introduction

Despite the remarkable progress of vision-language models (VLMs) in visual understanding through large-scale image-text alignment (Ghosh et al., 2024; Li et al., 2025b), current medical VLMs still frequently produce masks that contradict the accompanying text on key semantic attributes, especially in multi-site/multi-lesion scenarios. As illustrated in Fig. 1(a), even when the text explicitly specifies quantity and spatial cues such as "Bilateral", "two", or "upper left", the predicted masks may still fail to match the described number of lesions or their coarse locations. This phenomenon suggests that existing pipelines have not yet effectively transformed clinical language into pixel-level structural constraints, making it difficult to ensure text–mask consistency at the level of semantic attributes such as lesion count, laterality, and coarse spatial position.

We trace this mismatch to two underlying factors, as illustrated in Fig. 1(b). Firstly, clinical descriptions are highly templated and semantically repetitive, so the same short phrase can correspond to different imaging instances. In QaTa-COV19 dataset, for example, roughly 7,000 cases share only about 300 unique text templates, which means that the *same* text is frequently reused within the same training mini-batch rather than being a rare coincidence. Under this distribution, mainstream InfoNCE-style contrastive learning (Hu et al., 2024) still enforces a strict one-to-one matching, treating each image–text pair $(I^{(i)}, T^{(i)})$ as the only positive and all unpaired combinations $(I^{(i)}, T^{(j)}), j \neq i$ as negatives. In medical settings, however, $T^{(i)}$ and some $T^{(j)}$ can be exactly the same template (e.g., "unilateral pulmonary infection, one infected area"), so pairs like $(I^{(i)}, T^{(j)})$

---

*Corresponding author.

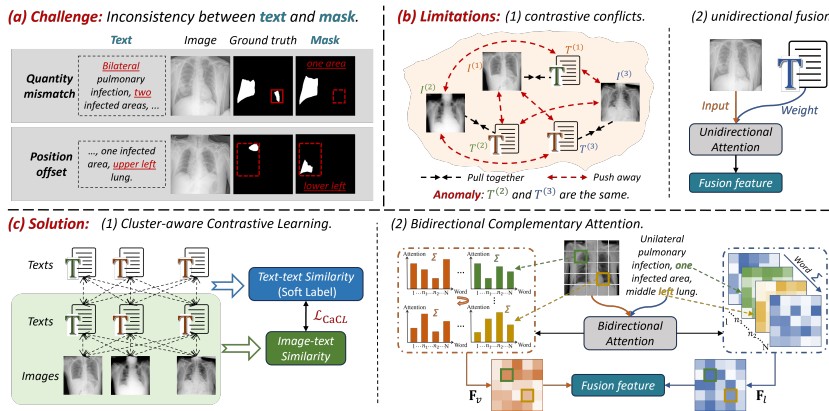

Figure 1: Challenge and limitations of existing methods, and our solution.

are incorrectly pushed apart as strong negatives. This hard-coding of nearly identical semantics as negatives produces a large number of false negatives and contrastive conflicts, ultimately degrading cross-modal alignment quality. Secondly, most existing methods still rely on vision-centric, unidirectional cross-attention mechanisms (Wang et al., 2024; Chen et al., 2025a). Although some works explore Dual-path attention (DualA) between vision and language in their fusion architectures (Huang et al., 2025), the textual features in these approaches typically only modulate visual features indirectly through attention weights, without forming an explicit, language-dominant spatial representation. As a result, such methods remain essentially vision-centric, and their exploitation and modeling of linguistic semantics are still insufficient.

To this end, we propose a Consistency-enhanced Two-stage Segmentation framework (**C2Seg**). As illustrated in Fig. 1(c), Stage I employs Cluster-aware Contrastive Learning (CaCL), which leverages implicit batch-wise semantic neighborhoods in a frozen language space to construct soft label distributions. This allows the image–text contrastive loss to fit a continuous semantic similarity distribution rather than hard positives/negatives, thereby alleviating contrastive conflicts and producing more discriminative, robustly aligned visual representations. In Stage II, we design a Bidirectional Complementary Attention Module (BCAM) that augments the traditional vision-dominant cross-attention with a language-dominant path while preserving spatial structure: the vision path outputs "each pixel enhanced by sentence-level semantics," whereas the language path captures "the aggregated per-token influence at each pixel," enabling spatially aware, deep bidirectional interaction. Furthermore, we introduce a KAN-based Attention Gating (K–Gate) that adaptively weights spatial locations and modality-specific features for fine-grained feature selection; KANs are also applied in the visual encoder and BCAM to provide nonlinear modeling capacity with limited parameter overhead. The contributions of this work can be summarized as follows:

- We propose CaCL that converts inter-text similarity into soft labels for contrastive learning, effectively suppressing false-negative conflicts, strengthening cross-modal alignment, and promoting semantic consistency.
- We design BCAM, which consists of two parallel vision-dominant and language-dominant paths, enabling deep bidirectional interaction and complementary enhancement while preserving spatial structure.
- We introduce K–Gate to perform nonlinear modeling of visual and language features separately through KANs, thereby achieving fine-grained selection of cross-modal information.
- We present C2Seg and demonstrate its significant improvements on text–mask consistency and segmentation accuracy through extensive experiments on four public medical datasets.

## 2 RELATED WORK

**Vision Language Models.** In recent years, the success of general vision-language pretraining models has driven multimodal research in the medical domain. For example, Zhang et al. (2025b)

performs image-text contrastive pretraining on large-scale biomedical figures and reports, learning generic medical image-text representations for retrieval and classification; Li et al. (2023) extends instruction-following VLMs to medical visual question answering and dialogue scenarios. Meanwhile, Cheng et al. (2023) and Ma et al. (2024) transfer the Segment Anything paradigm to medical imaging, achieving strong organ and lesion segmentation performance under prompts. Leveraging the rich multimodal semantic representations and scalable model interfaces offered by these foundation models, a number of vision-language guided segmentation methods have recently emerged(Tomar et al., 2022; Huemann et al., 2024; Li et al., 2024; Wang et al., 2025b; 2026). However, despite the diversity of fusion mechanisms, most existing multimodal segmentation models still rely on a unidirectional path where language guides vision, and lack explicit modeling of linguistic semantics. Although Liu et al. (2023); Cho et al. (2024); Sultan et al. (2025) construct bidirectional interaction structures between vision and language, their so-called "language-dominant" attention branches typically output only updated text tokens, while the final segmentation prediction still depends on feature maps from the visual branch. In other words, text mainly modulates visual features indirectly through attention weights, without forming an explicit language-centric spatial representation, which limits the modeling of fine-grained, text-conditioned spatial details.

**Contrastive Learning.** Multimodal pretraining networks utilize large-scale image-text pairs for contrastive learning to achieve effective cross-modal alignment (Mukhoti et al., 2023; Chng et al., 2024; Sung et al., 2024). In the context of medical image, the high cost of acquiring medical data and pixel-level annotations makes the incorporation of richly paired medical text and images a natural solution (Wang et al., 2022; Hu et al., 2023). Contrastive learning methods can align medical text and images, providing crucial semantic guidance for the segmentation process (Pan et al., 2025). However, contrastive objectives are known to suffer from false negatives and class collisions, and this issue is further exacerbated in clinical practice where medical reports are highly templated and heavily reused: the same or nearly identical wording can appear in many different cases, so traditional hard sampling often mislabels semantically similar pairs as negatives, compromising the stability and accuracy of the alignment. To mitigate such effects, Li et al. (2021) introduce cluster prototypes to encode semantic structure, while Dwibedi et al. (2021) augments positives with nearest neighbors in the feature space. Nonetheless, even when the contrastive objective is relaxed from strictly one-to-one to one-to-many, these methods still rely on hard positive/negative assignments at the label level, making it difficult to explicitly model the continuous semantic similarity distribution between samples and thus preventing them from effectively mitigating the soft-positive problem associated with semantic neighbors.

**Kolmogorov-Arnold Networks.** The Kolmogorov-Arnold representation theorem states that any multivariate continuous function $f(x_1, x_2, \ldots, x_n)$ can be represented as a finite composition of univariate continuous functions. Inspired by this, Liu et al. (2025) proposed the KAN, which replaces the linear weights in conventional MLPs with learnable univariate function units. This design enhances the model's nonlinear modeling capacity with fewer parameters and improves interpretability. Building on this idea, recent studies have introduced KANs into various vision tasks and achieved promising results (Chen et al., 2025b; Zhang et al., 2025a; Wang et al., 2025a; Zhu et al., 2026). Among them, Li et al. (2025a) integrated Tok-KAN blocks into the U-Net framework to strengthen local feature modeling, providing higher accuracy, efficiency and interpretability for vision tasks. Despite the demonstrated success of KANs in unimodal scenarios, its potential for cross-modal alignment and fusion has yet to be systematically explored.

## 3 METHOD

**Overview.** Our proposed C2Seg comprises two sequential stages, as illustrated in Fig. 2. Given input $I$ and $T$, we first design CaCL in the pretraining stage, where pairwise cosine similarity between text embeddings is computed to generate soft labels $Y_{ij}$ that guide robust image-text contrastive learning. In the fusion stage, we introduce BCAM, consisting of two paths that simultaneously generate vision-dominant features $\mathbf{F}_v$ and language-dominant features $\mathbf{F}_l$, enabling comprehensive exploitation of language information while preserving spatial structural information. Subsequently, K–Gate is applied to perform attention-weighted fusion of the two feature streams, producing the final output $\mathbf{F}_{\text{out}}$. Finally, through skip connections and upsampling operations, the spatial resolution is progressively restored, yielding high-quality pixel-level segmentation predictions. Notably, we

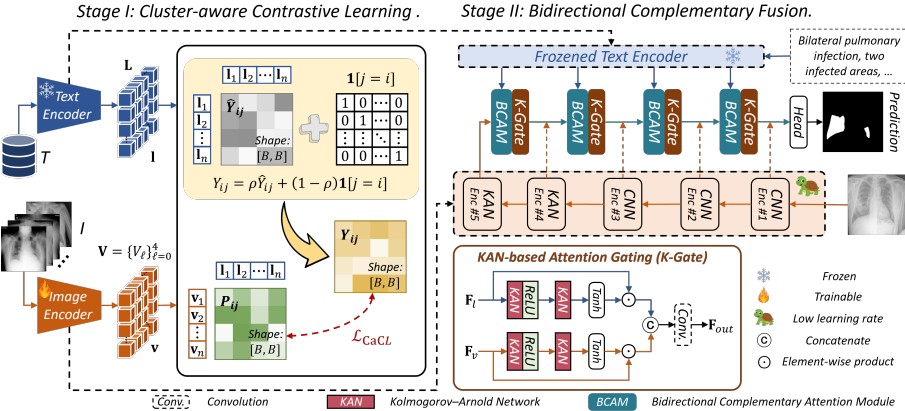

Figure 2: Overview of the proposed C2Seg. Given medical images and their corresponding textual descriptions, visual and language features are extracted using dedicated encoders.

fine-tune the visual encoder with a small learning rate while keeping the language encoder frozen to preserve stable semantic anchors in the text space.

## 3.1 FEATURE ENCODER

Based on the work of Li et al. (2025a) and Radford et al. (2021), we build a dual-branch encoder. The visual branch employs three convolutional layers followed by two KAN layers, aiming to combine the feature capturing ability of CNN with the nonlinear modeling capabilities of KANs. Given an image $I \in \mathbb{R}^{C \times H \times W}$, the visual branch produces multi-scale features $\mathbf{V} = \{V_\ell\}_{\ell=0}^4$ with $V_\ell \in \mathbb{R}^{C_\ell \times H_\ell \times W_\ell}$ and progressively downsampled spatial sizes $(H_\ell, W_\ell)$. A global image vector is computed as $\mathbf{v} = W_v \operatorname{GAP}(V_4) \in \mathbb{R}^{d_v}$, where $\operatorname{GAP}(\cdot)$ denotes global average pooling and $W_v$ is a learnable projection. On the language branch, the frozen CLIP encoder processes text $T$ tokenized to length $N$, yielding position-aware token embeddings $\mathbf{L} \in \mathbb{R}^{N \times d_t}$ and a sentence embedding $\mathbf{l} \in \mathbb{R}^{d_t}$. To enable channel-aligned bidirectional interaction at each visual scale, tokens are projected by learnable per-scale linear projections as $L_\ell = \mathbf{L} W_\ell$ with $W_\ell \in \mathbb{R}^{d_t \times C_\ell}$, hence $L_\ell \in \mathbb{R}^{N \times C_\ell}$. The set $\{V_\ell\}$ preserves pixel-level spatial structure for subsequent fusion and decoding, while $\{L_\ell\}$ together with $\mathbf{l}$ provide token-level and sentence-level semantics that feed into BCAM for bidirectional complementary interaction and fusion.

## 3.2 CLUSTER-AWARE CONTRASTIVE LEARNING

Unlike standard InfoNCE with a single positive/negative pairing, CaCL reframes in-batch contrastive learning as batch-wise semantic distribution matching. We first estimate text-text similarities in the frozen language space and convert them into soft labels, which are then used to supervise the image-text similarity distribution. This neighborhood-wise probabilistic supervision systematically suppresses erroneous repulsive gradients induced by templated clinical phrasing. Throughout, image embeddings $\{\mathbf{v}_i\}_{i=1}^B$ and sentence embeddings $\{\mathbf{l}_i\}_{i=1}^B$ are L2-normalized.

**Soft Label Construction.** Given a batch of size $B$, we first compute the text–text cosine similarity matrix $M_{ij} = \cos(\mathbf{l}_i, \mathbf{l}_j)$. To attenuate the global similarity inflation induced by shared templates, we apply row-mean debiasing and non-negativity clipping to each row, obtaining $M'_{ij} = \max\{M_{ij} - \mu_i, 0\}$, where $\mu_i = \frac{1}{B} \sum_k M_{ik}$. Here, $\mu_i$ can be viewed as a batch-level "template bias" whose removal helps suppress global template effects and yields a soft label distribution that focuses more on local semantic neighborhoods. Temperature $\tau$ is then used to produce semantic soft targets:

$$\hat{Y}_{ij} = \frac{\exp(M'_{ij}/\tau)}{\sum_k \exp(M'_{ik}/\tau)}. \tag{1}$$

To retain the anchor identity, the final target distribution mixes the diagonal with these soft targets as $Y_{ij} = \rho \hat{Y}_{ij} + (1-\rho)\,\mathbf{1}[j=i]$, where $\rho \in [0,1]$ balances the self-positive and its semantic neighbors.

**Distribution-supervised Symmetric InfoNCE.** Let the cross-modal logit be $s_{ij} = \mathbf{v}_i^\top \mathbf{l}_j$, and define the directional probabilities $P_{ij}^{v \to l} = \mathrm{softmax}_j(s_{ij}/\tau)$ and $P_{ij}^{l \to v} = \mathrm{softmax}_i(s_{ij}/\tau)$. We match the predicted distributions to $Y$ in both directions with a single bidirectional objective:

$$\mathcal{L}_{\mathrm{CaCL}} = -\frac{1}{B} \sum_{i=1}^{B} \sum_{j=1}^{B} \left[ Y_{ij} \log P_{ij}^{v \to l} + Y_{ji} \log P_{ij}^{l \to v} \right]. \tag{2}$$

**Theoretical Justification.** With the above definitions of $P^{v \to l}$ and $P^{l \to v}$, the gradient with respect to any logit $s_{ij}$ is

$$\frac{\partial \mathcal{L}_{\mathrm{CaCL}}}{\partial s_{ij}} = \frac{1}{B\tau} \left( P_{ij}^{v \to l} - Y_{ij} + P_{ij}^{l \to v} - Y_{ji} \right). \tag{3}$$

Semantically related but non-matching pairs thus receive nonzero target mass ($Y_{ij} > 0$ and/or $Y_{ji} > 0$), which attenuates or even reverses the repulsive gradient, mitigating false negatives and promoting neighborhood-consistent alignment. The additional computational cost is dominated by building $M$ in $O(B^2 C)$ time, which is negligible relative to the backbone.

### 3.3 BIDIRECTIONAL COMPLEMENTARY ATTENTION MODULE

Existing DualA-style bidirectional fusion methods typically update only the text tokens in their "language-dominant" branch, without producing features that explicitly preserve image spatial information. The representative work M3Att (Liu et al., 2023), although achieving bidirectional interaction between vision and language via multimodal cross-attention, uses a single fully connected projection to compress the spatial dimension $P$ into the channel dimension $C$ during fusion. This amounts to a non-structured mixing of all image patches for each text token, which weakens the original spatial inductive bias and tends to lose local details such as boundaries and textures. Therefore, we propose BCAM, which constructs two complementary attention paths in parallel during the fusion stage, one vision-dominant and one language-dominant, so that the two modalities can condition each other and produce spatially aligned multimodal features directly on the pixel grid. This alleviates modality imbalance and preserves more local information. The structures and differences of the three attention mechanisms are shown in Fig. 3.

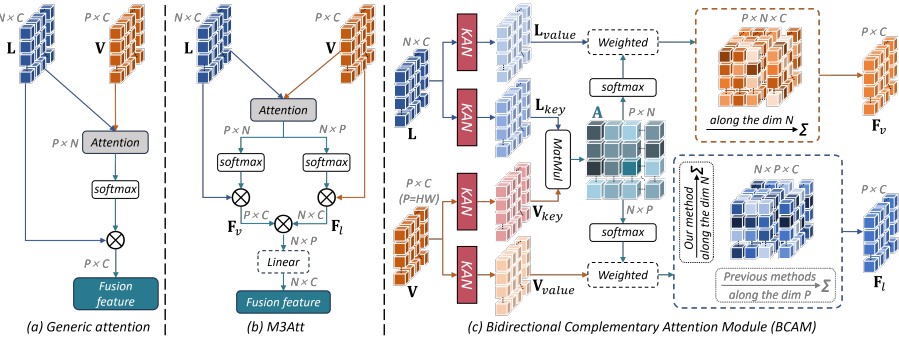

Figure 3: The architecture of (a) the generic attention mechanism, (b) the Multi-Modal Mutual Attention (M3Att), and (c) our proposed Bidirectional Complementary Attention Module (BCAM).

**Complementary Attention.** Given visual features $\mathbf{V} \in \mathbb{R}^{P \times C}$ ($P = H \times W$ denotes the number of spatial positions) and language features $\mathbf{L} \in \mathbb{R}^{N \times C}$, we use learnable KAN layers to obtain keys and values for cross-attention, defined as $\mathbf{V}_{\mathrm{key}} = \mathrm{KAN}(\mathbf{V})$, $\mathbf{V}_{\mathrm{value}} = \mathrm{KAN}(\mathbf{V})$, $\mathbf{L}_{\mathrm{key}} = \mathrm{KAN}(\mathbf{L})$, and $\mathbf{L}_{\mathrm{value}} = \mathrm{KAN}(\mathbf{L})$. We then construct the scaled dot-product attention scores $\mathbf{A}$:

$$\mathbf{A} = \frac{1}{\sqrt{d}} \mathbf{V}_{\mathrm{key}} \left( \mathbf{L}_{\mathrm{key}} \right)^\top \in \mathbb{R}^{P \times N}. \tag{4}$$

**Vision-dominant Path.** We apply a softmax along the language axis $N$ and use it to aggregate language values, yielding a language-enhanced visual representation:

$$\mathbf{F}_v = \text{Softmax}_N(\mathbf{A}) \cdot \mathbf{L}_{\text{value}} \in \mathbb{R}^{P \times C}. \tag{5}$$

The resulting $\mathbf{F}_v$ is a sentence- and token-injected visual feature map that preserves the original spatial resolution $P$. Intuitively, each pixel position collects token semantics according to its relevance weights, providing rich, pixel-level semantic context.

**Language-dominant Path.** Directly applying $\mathbf{A}^\top \in \mathbb{R}^{N \times P}$ to $\mathbf{V}_{\text{value}}$ would produce a token-level representation of shape $\mathbb{R}^{N \times C}$, which lacks explicit pixel topology and thus hampers subsequent decoding. Therefore, we normalize the transpose $\mathbf{A}^\top$ along the spatial axis $P$, apply each token's spatial weights to the visual values, and then aggregate across tokens to obtain a language-guided feature aligned with the image grid:

$$\mathbf{F}_l = \frac{1}{N} \sum_{n=1}^{N} \text{Softmax}_P(\mathbf{A}^\top[n, :]) \odot \mathbf{V}_{\text{value}} \in \mathbb{R}^{P \times C}, \tag{6}$$

where $\odot$ denotes broadcasted element-wise multiplication over the channel dimension. The resulting $\mathbf{F}_l$ encodes each token's collective attention over all spatial locations, forming a spatially coherent, language-guided feature map. Compared with a unidirectional, vision-only fusion scheme, the addition of this language-initiated path mitigates modality imbalance, improves cross-modal alignment, and provides spatially consistent signals to the decoder, thereby enhancing downstream segmentation performance.

### 3.4 KAN-BASED ATTENTION GATING

After the BCAM, the visual and textual streams have exchanged information bidirectionally; however, modality-specific statistical biases and noise patterns (e.g., imaging artifacts and templated phrasing) may still propagate across modalities and be amplified, thereby diluting spatial detail or inducing semantic drift. To address this, we introduce a KAN-based nonlinear gating mechanism that performs selective suppression and enhancement within each modality before fusion, followed by data-dependent mixing. This design improves the expressiveness of features and alleviates modal imbalance from the source.

For the two feature streams output by BCAM, the visual branch $\mathbf{F}_v \in \mathbb{R}^{P \times C}$ and the language branch $\mathbf{F}_l \in \mathbb{R}^{P \times C}$, we construct two independent KAN gating heads, each comprising two KAN layers with an intermediate ReLU and producing a gating tensor that matches the input shape. We adopt a $\tanh$ activation to normalize each branch output, bounding the learned gating weights to $[-1, 1]$. Accordingly, the gating tensors are defined as:

$$\mathbf{g}_v = \tanh\big(\text{KAN}_v^{(2)}(\text{ReLU}(\text{KAN}_v^{(1)}(\mathbf{F}_v)))\big), \quad \mathbf{g}_l = \tanh\big(\text{KAN}_l^{(2)}(\text{ReLU}(\text{KAN}_l^{(1)}(\mathbf{F}_l)))\big). \tag{7}$$

We then perform element-wise reweighting $\mathbf{F}_v^g = \mathbf{F}_v \odot \mathbf{g}_v$ and $\mathbf{F}_l^g = \mathbf{F}_l \odot \mathbf{g}_l$, where $\odot$ denotes element-wise multiplication. After this modality-internal refinement, we concatenate $[\mathbf{F}_v^g \| \mathbf{F}_l^g]$ along the channel dimension and apply a $1 \times 1$ convolution for channel alignment and linear mixing to obtain the final fused representation $\mathbf{F}_{\text{out}}$. This process enhances the spatial perception ability of the model through feature selection, while providing a nonlinear and fine-grained selective path for cross-modal fusion.

## 4 EXPERIMENTS

### 4.1 DATASETS AND IMPLEMENTATION DETAILS

**Datasets.** We conduct experiments on four public medical image segmentation datasets: QaTa-COV19 (Degerli et al., 2022), MosMedData+ (Morozov et al., 2020; Hofmanninger et al., 2020),

Table 1: Quantitative comparison on QaTa-COV19 and MosMedData+ datasets.

| Method | Params(M) | Text | QaTa-COV19 | | | | MosMedData+ | | | |
|---|---|---|---|---|---|---|---|---|---|---|
| | | | Dice(%)↑ | mIoU(%)↑ | HD95↓ | ASSD↓ | Dice(%)↑ | mIoU(%)↑ | HD95↓ | ASSD↓ |
| U-Net (Ronneberger et al., 2015) | 31.4 | ✗ | 79.02 | 69.46 | 33.98 | 9.03 | 64.60 | 50.73 | 23.52 | 6.35 |
| U-Net++ (Zhou et al., 2018) | 74.5 | ✗ | 79.62 | 70.25 | 36.14 | 9.91 | 71.75 | 58.39 | 24.06 | 5.45 |
| nnUNet (Isensee et al., 2021) | 19.1 | ✗ | 80.42 | 70.81 | 28.14 | 9.86 | 72.59 | 60.36 | 22.75 | 5.56 |
| Swin-Unet (Cao et al., 2023) | 82.3 | ✗ | 78.07 | 68.34 | 31.51 | 9.20 | 63.29 | 50.19 | 25.31 | 7.69 |
| TransUNet (Chen et al., 2024) | 105.0 | ✗ | 78.63 | 69.13 | 29.88 | 8.42 | 71.24 | 58.44 | 23.41 | 6.38 |
| UKAN (Li et al., 2025a) | **9.4** | ✗ | 79.30 | 69.85 | 31.89 | 8.79 | 72.56 | 59.05 | 29.38 | 7.25 |
| MM-UKAN++ (Zhang et al., 2025a) | 9.9 | ✗ | 79.20 | 69.70 | 35.26 | 9.76 | 71.82 | 58.37 | 32.63 | 8.96 |
| CLIP (Radford et al., 2021) | 87.0 | ✓ | 79.81 | 70.66 | 23.25 | 5.54 | 71.97 | 59.64 | 26.24 | 6.58 |
| GLoRIA (Huang et al., 2021) | 45.6 | ✓ | 79.94 | 70.68 | 26.47 | 5.24 | 72.42 | 60.18 | 28.61 | 6.79 |
| ViLT (Kim et al., 2021) | 87.4 | ✓ | 79.63 | 70.12 | 25.32 | 5.96 | 72.36 | 60.15 | 24.85 | 5.69 |
| TGANet (Tomar et al., 2022) | 19.8 | ✓ | 77.17 | 64.39 | 29.54 | 7.83 | 69.48 | 55.81 | 26.39 | 6.12 |
| ConVIRT (Zhang et al., 2022) | 35.2 | ✓ | 79.72 | 70.58 | 22.36 | 6.03 | 72.06 | 59.73 | 22.38 | 6.36 |
| LAVT (Yang et al., 2022) | 118.6 | ✓ | 80.48 | 67.01 | 15.70 | 4.87 | 68.51 | 55.32 | 17.28 | 4.18 |
| SLViT (Ouyang et al., 2023) | 131.5 | ✓ | 79.25 | 68.87 | 15.18 | 4.35 | 72.57 | 60.78 | 21.23 | 6.10 |
| LViT (Li et al., 2024) | 29.7 | ✓ | 81.52 | 68.63 | 18.62 | 5.32 | 72.10 | 57.35 | 18.94 | 4.82 |
| UniLSeg (Liu et al., 2024) | 28.7 | ✓ | 72.88 | 59.58 | 15.15 | 4.11 | 65.89 | 52.01 | 19.98 | 4.96 |
| RefSegformer (Wu et al., 2024) | 195.0 | ✓ | 81.63 | 69.71 | 20.22 | 5.29 | 70.25 | 57.31 | 19.70 | 4.78 |
| MedLangViT (Wang et al., 2025b) | 27.7 | ✓ | 84.27 | 75.93 | 14.51 | 3.97 | 75.95 | 63.17 | 18.29 | 4.12 |
| ARSeg (Wang et al., 2026) | 30.1 | ✓ | 84.09 | 72.64 | 19.90 | 5.24 | 73.24 | 59.82 | 31.88 | 7.65 |
| **C2Seg (Ours)** | 18.92 | ✓ | **85.25** | **76.97** | **12.71** | **3.38** | **77.81** | **65.17** | **15.02** | **3.76** |

CVC-ClinicDB (Bernal et al., 2015), and Kvasir (Jha et al., 2020). QaTa-COV19 and MosMedData+ are lung datasets with paired textual annotations provided by Li et al. (2024), while CVC-ClinicDB and Kvasir are colon polyp datasets whose image-text pairs are constructed following Zhang et al. (2024). We adopt Dice, mean Intersection over Union (mIoU), the 95th percentile Hausdorff Distance (HD95), and the Average Symmetric Surface Distance (ASSD) as our primary evaluation metrics. Detailed descriptions of these datasets and the evaluation metrics are given in Appendix D and Appendix E, respectively.

**Implementation Details.** Our proposed method is implemented on a server with two NVIDIA GeForce RTX 5090 D (32 GB) GPUs. By default, the batch size is set to 256 for the pretraining stage and 32 for the segmentation stage, the temperature parameter $\tau$ is set to 0.07, and the soft-label weight parameter $\rho$ is set to 0.8. During the supervised segmentation stage, we adopt a combined BCE + Dice objective (BCEDice loss) as the training loss; a detailed formulation of this loss is provided in Appendix F. The initial learning rate is set to $1 \times 10^{-3}$ for QaTa-COV19 and CVC-ClinicDB, and $3 \times 10^{-4}$ for MosMedData+ and Kvasir. We use the Adam optimizer together with a cosine annealing learning rate scheduler.

Table 2: Quantitative comparison on CVC-ClinicDB and Kvasir datasets.

| Method | Params(M) | Text | CVC-ClinicDB | | | | Kvasir | | | |
|---|---|---|---|---|---|---|---|---|---|---|
| | | | Dice(%)↑ | mIoU(%)↑ | HD95↓ | ASSD↓ | Dice(%)↑ | mIoU(%)↑ | HD95↓ | ASSD↓ |
| U-Net (Ronneberger et al., 2015) | 31.4 | ✗ | 57.57 | 44.93 | 49.40 | 19.87 | 75.77 | 65.20 | 40.29 | 12.11 |
| U-Net++ (Zhou et al., 2018) | 74.5 | ✗ | 88.94 | 82.91 | 12.16 | 3.99 | 87.00 | 79.71 | 20.87 | 6.63 |
| nnUNet (Isensee et al., 2021) | 19.1 | ✗ | 85.69 | 77.72 | 13.48 | 5.84 | 86.95 | 79.19 | 20.39 | 5.81 |
| Swin-Unet (Cao et al., 2023) | 82.3 | ✗ | 81.19 | 71.64 | 26.38 | 8.67 | 77.24 | 66.90 | 21.25 | 8.80 |
| UKAN (Li et al., 2025a) | **9.4** | ✗ | 89.74 | 84.47 | 13.19 | 3.72 | 87.77 | 81.13 | 21.37 | 5.82 |
| MM-UKAN++ (Zhang et al., 2025a) | 9.9 | ✗ | 89.52 | 82.15 | 13.26 | 3.36 | 85.63 | 78.03 | 24.81 | 7.12 |
| LAVT (Yang et al., 2022) | 118.6 | ✓ | 88.13 | 82.76 | 9.33 | 3.85 | 90.83 | 84.90 | 15.90 | 4.15 |
| TGANet (Tomar et al., 2022) | 19.8 | ✓ | 89.93 | 84.56 | 8.41 | **2.11** | 90.44 | 84.19 | 17.18 | 4.33 |
| SLViT (Ouyang et al., 2023) | 131.5 | ✓ | 80.55 | 72.86 | 26.91 | 9.87 | 85.69 | 77.97 | 19.57 | 5.56 |
| LViT (Li et al., 2024) | 29.7 | ✓ | 88.27 | 80.81 | 15.18 | 4.15 | 87.36 | 79.85 | 24.18 | 6.40 |
| RefSegformer (Wu et al., 2024) | 195.0 | ✓ | 80.73 | 71.94 | 23.10 | 7.28 | 86.87 | 78.77 | 25.30 | 6.46 |
| MMIUNet (Bui et al., 2024) | 56.2 | ✓ | 89.96 | 84.14 | 11.55 | 4.02 | 90.27 | 84.29 | 15.13 | 4.35 |
| RecLMIS (Huang et al., 2025) | 23.7 | ✓ | 81.31 | 73.04 | 38.44 | 12.13 | 90.63 | 84.35 | 17.93 | 4.43 |
| MedLangViT (Wang et al., 2025b) | 27.7 | ✓ | 88.35 | 81.92 | 10.66 | 4.50 | 90.57 | 84.21 | 14.12 | 4.09 |
| ARSeg (Wang et al., 2026) | 30.1 | ✓ | 89.74 | 82.64 | 13.71 | 4.29 | 88.45 | 81.34 | 22.79 | 5.66 |
| **C2Seg (Ours)** | 18.92 | ✓ | **91.82** | **86.81** | **6.53** | 2.23 | **91.92** | **85.27** | **13.62** | **3.98** |

## 4.2 PERFORMANCE COMPARISON

We conduct a systematic comparison between C2Seg and other segmentation methods on four public medical image datasets, with quantitative results summarized in Tab. 1 and Tab. 2. Overall, C2Seg achieves competitive or even state-of-the-art performance across all four benchmarks. Benefiting from the incorporation of textual modality, C2Seg significantly outperforms the strongest CNN-based model nnU-Net and the KAN-based model UKAN. Compared with other multimodal approaches, our method not only excels on region-overlap metrics (Dice and mIoU), but also shows clear advantages on distance-based metrics HD95 and ASSD, indicating that C2Seg provides more

accurate delineation of lesion locations and boundaries and, from a geometric perspective, further corroborating the benefit of leveraging textual constraints for spatial localization.

As illustrated in Fig. 4, we further analyze the behavior of different methods. In the first row, the text explicitly specifies "unilateral pulmonary infection, one infected area", yet several methods still misclassify regions in the right lung as lesions, whereas C2Seg correctly restricts its prediction to a single lesion in the left lung. In the second row, the textual prompt "middle lower left lung" is given; some methods fail to properly capture positional information and produce over-segmentation in the upper left lung, while C2Seg more accurately focuses on the middle-lower region of the left lung, demonstrating finer text-guided localization. In the fifth row, the text includes a numerical cue "two infected areas"; some methods produce an incorrect number of lesions, whereas C2Seg matches both the spatial locations and the number of lesions described in the text. These qualitative observations are consistent with the quantitative gains and further confirm the advantage of C2Seg in adhering to textual constraints and improving text–mask consistency.

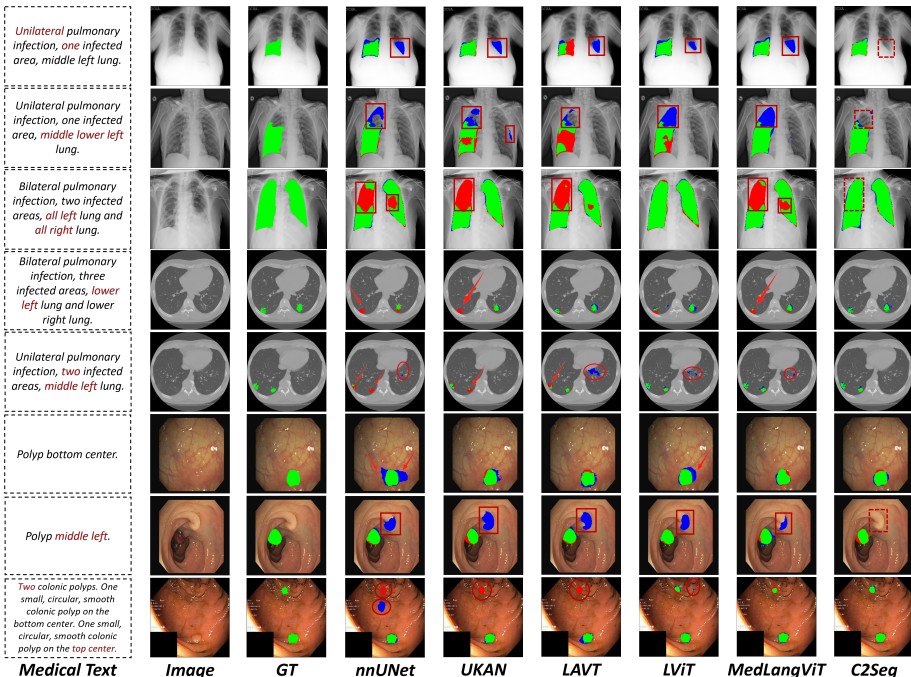

Figure 4: Visualization of different methods. Green, red, and blue represent true positive, false negative, and false positive pixels, respectively.

## 4.3 ABLATION STUDIES

**Effect of Proposed Components.** We conduct systematic ablation studies on the MosMedData+ and CVC-ClinicDB datasets, as summarized in Tab. 3. Case (a) keeps only the visual branch as a unimodal baseline; Case (b)-(f) all use CLIP as the text encoder, where (b) adopts DualA in Stage II, (c) replaces DualA with BCAM, (d) further adds K-Gate on top of BCAM, (e) introduces conventional hard-label contrastive learning (HardCL) in Stage I, and the final full model corresponds to Case (f). The results show a clear step-wise improvement as components are added, validating both the effectiveness and complementarity of the proposed modules.

**Effect of Text Encoders.** Keeping the training framework fixed, we further replace the text encoder with domain-specific biomedical language models BioBERT-Base v1.1 (Lee et al., 2020) and PubMedBERT (Gu et al., 2021). As shown in Tab. 3, even though CLIP text encoder is not pretrained on medical corpora, its performance remains slightly better than BioBERT and PubMed-BERT. We speculate this is mainly because the CLIP is learned via contrastive training on large-scale

Table 3: Ablation study on MosMedData+ and CVC-ClinicDB datasets.

| Case | Text encoder | Stage I | | Stage II | | | MosMedData+ | | CVC-ClinicDB | |
|------|--------------|---------|------|----------|------|--------|-------------|-----------|--------------|-----------|
| | | HardCL | CaCL | DualA | BCAM | K-Gate | Dice(%)↑ | mIoU(%)↑ | Dice(%)↑ | mIoU(%)↑ |
| (a) | × | | | | | | 73.61 | 60.02 | 86.59 | 78.11 |
| (b) | | | | ✓ | | | 75.59 | 62.35 | 89.56 | 83.14 |
| (c) | CLIP | | | ✓ | ✓ | | 76.50 | 63.77 | 90.31 | 84.45 |
| (d) | | | | ✓ | ✓ | ✓ | 77.03 | 64.45 | 90.68 | 85.98 |
| (e) | | ✓ | | ✓ | ✓ | ✓ | 77.32 | 64.51 | 91.26 | 86.33 |
| (f) | | | ✓ | ✓ | ✓ | ✓ | **77.81** | **65.17** | **91.82** | **86.81** |
| (g) | BioBERT-Base | | ✓ | ✓ | ✓ | ✓ | 76.22 | 64.18 | 90.60 | 86.08 |
| (h) | PubMedBERT | | ✓ | ✓ | ✓ | ✓ | 76.69 | 64.76 | 90.25 | 84.69 |

image-text pairs, yielding an embedding space that is naturally aligned with visual features and better matched to the short, location- and quantity-oriented descriptions used in this work, whereas BioBERT/PubMedBERT are pretrained solely on pure-text biomedical corpora and are better suited for long clinical reports, lacking such cross-modal alignment and adaptation.

**Effect of Bidirectional Fusion Mechanism.** To verify the effectiveness of the proposed bidirectional complementary attention, we fix the backbone and decoder and only replace the Stage II fusion module, comparing DualA with the proposed BCAM. The visualization results on QaTa-COV19 and Kvasir datasets are shown in Fig. 5. Even when the prompt explicitly contains quantitative and positional cues such as "Unilateral," "two," and "lower left," DualA still tends to produce results that deviate from the text, for example activating both lungs or merging multiple lesions into a single region. In contrast, with BCAM, the high-response regions are much better concentrated on the text-specified target areas. This indicates that the language-dominant path effectively strengthens the mapping from textual semantics to spatial representations and, when combined with the vision-dominant path, not only alleviates text–mask inconsistency but also leads to more accurate lesion localization and boundary delineation.

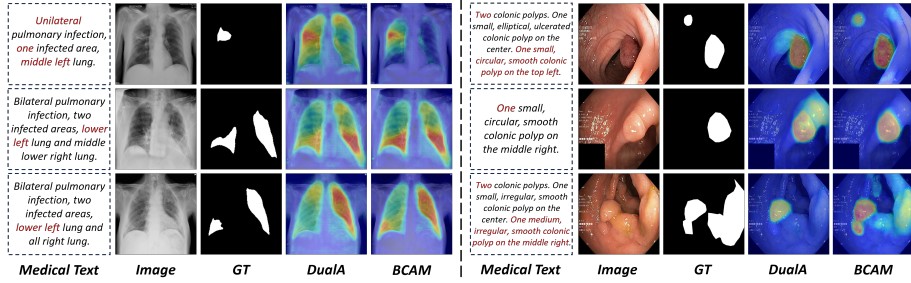

Figure 5: Qualitative comparison of DualA and BCAM on QaTa-COV19 and Kvasir datasets.

**Effect of KAN Structure.** To assess the role of KAN in our framework, we design five structural variants on the MosMedData+ dataset: using a pure CNN as the visual encoder; adopting a hybrid encoder with the first three layers as CNN and the last two layers as KAN; replacing the visual encoder entirely with a pure KAN; replacing the KAN layers in BCAM with linear projections while keeping the encoder unchanged; and replacing the KAN layers in K–Gate with standard MLPs. As shown in Tab. 4(a), the hybrid encoder yields consistent performance gains over the pure CNN encoder, whereas the pure KAN encoder even leads to degradation, indicating that the local receptive fields provided by CNNs remain crucial for low-level feature extraction and that KAN is more suitable as a high-level complement rather than a full replacement. In addition, substituting KAN with linear layers or MLPs in BCAM or K–Gate results in performance drops, suggesting that the nonlinear modeling capacity of KAN still offers advantages for higher-order cross-modal interactions.

**Effect of Hyperparameters.** We conduct a hyperparameter sensitivity analysis on the MosMed-Data+ dataset, with the results summarized in Tab. 4(b). For each experiment, all settings are fixed to the default configuration ($\rho$=0.8, $\tau$=0.07, batch size=256) except for the hyperparameter under

Table 4: Ablation of KAN and hyperparameters on MosMedData+ datasets.

| (a) Ablations on KAN. | | | (b) Sensitivity of hyperparameters. | | | | | | | | |
|---|---|---|---|---|---|---|---|---|---|---|---|
| Settings | Dice(%)↑ | mIoU(%)↑ | Effect of $\rho$ | | | Effect of $\tau$ | | | Effect of batch size | | |
| CNN encoder | 77.29 | 64.51 | ($\tau = 0.07$, batch size = 256) | | | ($\rho = 0.8$, batch size = 256) | | | ($\rho = 0.8$, $\tau = 0.07$) | | |
| Hybrid encoder | **77.81** | **65.17** | $\rho$ | Dice(%)↑ | mIoU(%)↑ | $\tau$ | Dice(%)↑ | mIoU(%)↑ | batch size | Dice(%)↑ | mIoU(%)↑ |
| KAN encoder | 76.21 | 63.27 | 0.6 | 77.64 | 65.05 | 0.05 | 77.59 | 65.12 | 128 | 76.52 | 63.93 |
| BCAM (Linear) | 77.26 | 64.42 | 0.8 | **77.81** | **65.17** | 0.07 | **77.81** | **65.17** | 256 | 77.81 | 65.17 |
| K-Gate (MLP) | 76.93 | 64.02 | 1.0 | 77.12 | 64.26 | 0.10 | 77.45 | 64.98 | 512 | **77.89** | **65.18** |

investigation. The results show that the model is only weakly sensitive to $\rho$ and $\tau$, and we therefore adopt the slightly better and more stable combination $\rho$=0.8 and $\tau$=0.07 as the default setting. In addition, as the batch size increases, the evaluation metrics exhibit a general upward trend, confirming the positive effect of larger batches on contrastive learning; however, the gain of 512 over 256 is marginal, so we choose batch size=256 in the main experiments as a trade-off between performance and computational cost.

**Effect of Contrastive Learning Strategy.** We compare CaCL with two control settings (without contrastive learning and with generic contrastive learning) on the QaTa-COV19 and MosMedData+ datasets. For each dataset, we select the top 5 textual descriptions with the largest number of matched images and their corresponding image samples, and visualize the distributions of their image features in the latent space under different training strategies. As shown in Fig. 6, CaCL yields more compact and semantically better separated cross-modal features, confirming its effectiveness in improving cross-modal alignment.

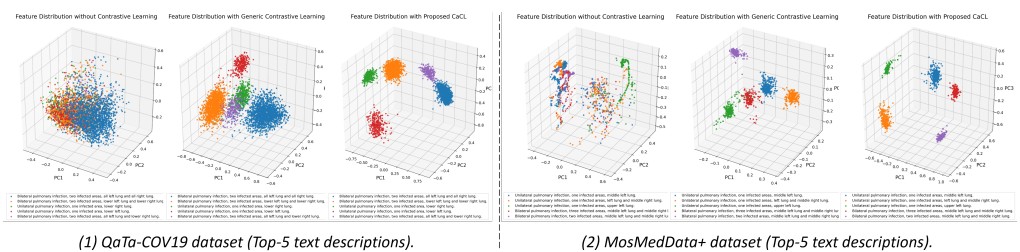

*(1) QaTa-COV19 dataset (Top-5 text descriptions).*  *(2) MosMedData+ dataset (Top-5 text descriptions).*

Figure 6: Visualization of image feature distribution on QaTa-COV19 and MosMedData+ datasets under three pretraining strategies (no contrast, general contrast, and CaCL). Each color represents all images corresponding to a textual description.

## 5 CONCLUSION

In this paper, we propose C2Seg, a two-stage medical image segmentation framework targeting text-mask consistency. In the pretraining stage, we first reformulate batch contrastive learning as cluster-level distribution matching, transforming text-to-text similarity into a soft target to suppress false negatives and stabilize cross-modal alignment. In the fusion stage, we establish a modality fusion path dominated by vision and language to more fully exploit textual information, while K–Gate provides fine-grained cross-modal information selection. Extensive experiments on four challenging datasets validate the superiority and effectiveness of C2Seg.

## 6 LIMITATIONS

In this work, we primarily rely on qualitative visualization to assess text–mask consistency and have not yet introduced a dedicated quantitative metric for this purpose. Moreover, the textual descriptions in our datasets are relatively clean and structured, so we have not systematically examined the model's robustness to complex text noise, such as spelling errors, abbreviations, or missing findings. Both aspects are important directions that we plan to explore in future work.

ACKNOWLEDGMENTS

This work was supported in part by the Hefei Municipal Natural Science Foundation (2022009) and the Jiangsu Provincial Association of Traditional Chinese Medicine Project (CYTF2024089). Also, the authors gratefully acknowledge the High-performance Computing Platform of Anhui University for providing computing resources.

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

## A    ETHICS STATEMENT

We have ensured that all data used in this research comply with ethical guidelines and proper citations. The study follows responsible AI practices and avoids any data manipulation or misrepresentation.

## B    REPRODUCIBILITY STATEMENT

The experiments in this work are fully reproducible. All datasets and code used for model training and evaluation are publicly available. Detailed information on the experimental setup, model configurations, and hyperparameters is provided to enable independent reproduction of results.

## C    USE OF LLMS

This work does not involve the use of any large language models (LLMs), such as GPT, DeepSeek, or similar models.

## D    DATASETS

To comprehensively evaluate the effectiveness of our proposed C2Seg, we conduct experiments on four widely used public medical image segmentation datasets: QaTa-COV19, MosMedData+, CVC-ClinicDB, and Kvasir. QaTa-COV19 and MosMedData+ are chest imaging datasets, both augmented with textual descriptions curated by Li et al. (2024), which provide descriptive information for each image. A notable challenge in these two datasets is the high degree of text reuse: for example, over 7,000 samples in QaTa-COV19 share only about 300 different textual descriptions, which poses additional difficulties for fine-grained vision–language alignment. In addition, we consider two colonoscopy polyp segmentation datasets, CVC-ClinicDB and Kvasir, where image–text pairs are constructed by Zhang et al. (2024) through clinically oriented descriptions of each image.

The QaTa-COV19 dataset Degerli et al. (2022), jointly released by Qatar University and Tampere University, comprises 9,258 chest X-ray images with pixel-level annotations for COVID-19 lesion segmentation. We follow the dataset partition protocol used in LViT Li et al. (2024), splitting the dataset into a training set (5,716 images), a validation set (1,429 images), and a test set (2,113 images).

The MosMedData+ dataset Morozov et al. (2020); Hofmanninger et al. (2020) includes 2,729 chest CT slices labeled with infection masks for pulmonary lesions. Each image is also accompanied by textual descriptions that specify attributes such as infection laterality, anatomical location, and affected lung lobes, providing diverse spatial and semantic cues. We adopt the same data split as in LViT Li et al. (2024): 2,183 training images, 273 validation images, and 273 test images.

The CVC-ClinicDB dataset Bernal et al. (2015) contains 612 colonoscopy images with pixel-level polyp masks collected from 29 colonoscopy video sequences. In this work, we use the text-augmented version released by Zhang et al. (2024), where each image is associated with a concise clinical-style description. Following Zhang et al. (2024), we split CVC-ClinicDB into 550 training images and 62 test images.

The Kvasir dataset Jha et al. (2020) consists of 1,000 colonoscopy images with expert-annotated polyp masks acquired under diverse clinical conditions. Similarly, we adopt the text annotations provided by Zhang et al. (2024) and follow its partition protocol, using 900 images for training and 100 images for testing.

## E    EVALUATION METRICS

To quantitatively assess segmentation performance, we adopt four widely used evaluation metrics: Dice score, mean Intersection over Union (mIoU), the 95th percentile Hausdorff distance (HD95), and the Average Symmetric Surface Distance (ASSD). Dice and mIoU evaluate the region-wise

overlap between the predicted segmentation and ground-truth annotations across all categories and spatial positions.

The Dice score measures the harmonic mean of precision and recall, placing more emphasis on correctly segmented regions, and is defined as:

$$\text{Dice} = \sum_{i=1}^{N} \sum_{j=1}^{C} \frac{1}{NC} \cdot \frac{2|p_{ij} \cap y_{ij}|}{|p_{ij}| + |y_{ij}|}, \tag{8}$$

where $p_{ij}$ and $y_{ij}$ denote the predicted and ground-truth pixel sets, respectively, for sample $i$ and class $j$.

The mean Intersection over Union (mIoU) reflects the average segmentation accuracy by computing the ratio of intersection over union for each class and averaging over the dataset:

$$\text{mIoU} = \sum_{i=1}^{N} \sum_{j=1}^{C} \frac{1}{NC} \cdot \frac{|p_{ij} \cap y_{ij}|}{|p_{ij} \cup y_{ij}|}, \tag{9}$$

where $N$ denotes the number of samples and $C$ the total number of semantic categories. The notation $|p_{ij} \cap y_{ij}|$ and $|p_{ij} \cup y_{ij}|$ indicate the number of overlapping and combined pixels, respectively, between prediction and ground truth for each sample–category pair.

In addition to region-overlap metrics, we also report two boundary-based distance metrics, HD95 and ASSD, to assess the accuracy of lesion localization and contour delineation. Let $S_{\text{pred}}$ and $S_{\text{gt}}$ denote the sets of surface (boundary) points of the predicted and ground-truth segmentations, respectively, and $d(x, S) = \min_{y \in S} \|x - y\|$ be the shortest distance from a point $x$ to a surface $S$. We first compute all directed surface distances $\{d(x, S_{\text{gt}}) \mid x \in S_{\text{pred}}\}$ and $\{d(y, S_{\text{pred}}) \mid y \in S_{\text{gt}}\}$ and take their union. HD95 is then defined as the 95th percentile of this pooled distance distribution, i.e., the value below which 95% of all surface distances lie. A lower HD95 indicates that even the worst-aligned boundary regions are relatively close to the ground truth.

The Average Symmetric Surface Distance (ASSD) measures the mean bidirectional surface distance between prediction and ground truth:

$$\text{ASSD}(S_{\text{pred}}, S_{\text{gt}}) = \frac{1}{2} \left( \frac{1}{|S_{\text{pred}}|} \sum_{x \in S_{\text{pred}}} d(x, S_{\text{gt}}) + \frac{1}{|S_{\text{gt}}|} \sum_{y \in S_{\text{gt}}} d(y, S_{\text{pred}}) \right). \tag{10}$$

Smaller ASSD values indicate that, on average, the predicted and ground-truth boundaries are closely aligned in space. Together with Dice and mIoU, HD95 and ASSD provide a more comprehensive evaluation of both region overlap and boundary geometry.

## F  TRAINING OBJECTIVE IN STAGE II

In Stage II, the segmentation decoder is optimized with a hybrid Binary Cross-Entropy (BCE) and Dice loss, which combines stable pixel-wise supervision with region-overlap awareness. Let $p_i \in [0, 1]$ denote the predicted foreground probability for pixel $i$ and $y_i \in \{0, 1\}$ the corresponding ground-truth label (foreground or background), over all pixels indexed by $i = 1, \ldots, |\Omega|$ in the image domain $\Omega$.

The BCE loss is defined as

$$\mathcal{L}_{\text{BCE}} = -\frac{1}{|\Omega|} \sum_{i=1}^{|\Omega|} \left[ y_i \log(p_i) + (1 - y_i) \log(1 - p_i) \right], \tag{11}$$

which treats segmentation as a pixel-wise binary classification task and provides numerically stable gradients, especially in the early training phase.

To explicitly encourage region-level overlap between the prediction and ground truth and alleviate class imbalance, we further adopt a soft Dice loss:

$$\mathcal{L}_{\text{Dice}} = 1 - \frac{2\sum_{i=1}^{|\Omega|} p_i y_i}{\sum_{i=1}^{|\Omega|} p_i + \sum_{i=1}^{|\Omega|} y_i + \epsilon}, \tag{12}$$

where $\epsilon$ is a small constant to avoid division by zero. This term focuses on the agreement between predicted and ground-truth foreground regions and is particularly beneficial when the lesion occupies only a small portion of the image.

The final segmentation loss in Stage II is a weighted combination of the two terms:

$$\mathcal{L}_{\text{seg}} = 0.5\,\mathcal{L}_{\text{BCE}} + 0.5\,\mathcal{L}_{\text{Dice}}. \tag{13}$$

In practice, this simple 1:1 weighting leverages the complementary strengths of BCE and Dice: the former stabilizes optimization at the pixel level, while the latter improves the overlap quality of the predicted masks, which is crucial for accurate medical image segmentation.

