# OpenReview forum: "Towards Text-Mask Consistency in Medical Image Segmentation"
_ICLR.cc/2026/Conference — ICLR 2026 Poster_

### Official Review · Reviewer_RBwK · 2025-10-26

**Soundness:** 2
**Presentation:** 3
**Contribution:** 2
**Rating:** 4
**Confidence:** 4

**Summary:**

This paper proposes C2Seg, a vision-language framework for medical image segmentation that aims to achieve better alignment between clinical text descriptions and visual features. It introduces a Cluster-aware Contrastive Learning (CaCL) method that uses soft labels derived from text–text similarity in a frozen language space to address issues of repetitive and templated clinical phrasing, which can create false negatives in traditional contrastive setups. To fuse the two modalities, the paper presents a Bidirectional Complementary Attention Module (BCAM), which performs dual cross-attention between image and text tokens while preserving spatial structure. Finally, a KAN-based nonlinear gating mechanism (K-Gate) is added to selectively suppress or enhance modality-specific noise before fusion. Experiments on multiple medical segmentation datasets show moderate improvements in Dice and mIoU scores, suggesting that the proposed modules enhance cross-modal alignment and segmentation quality.

**Strengths:**

The main strength of the paper lies in its idea of using soft labels for contrastive learning through the proposed Cluster-aware Contrastive Learning (CaCL) module. Instead of relying on traditional hard negatives, which can misclassify semantically similar clinical descriptions as distinct, the method introduces a more flexible alignment strategy by estimating text–text similarity in a frozen language space and converting it into soft supervision. This is a smart and well-motivated way to handle the repetitive and templated nature of clinical text, and it effectively reduces false negatives during image–text alignment. The paper is also well-presented, with clear and visually appealing illustrations that help convey the framework and intuition behind each component. In addition, the authors conduct extensive quantitative experiments and compare their method against a wide range of state-of-the-art models, showing consistent, if moderate, performance improvements, which supports the effectiveness of their overall approach.

**Weaknesses:**

Although Figure 1 looks visually good, I think it is a bit overcomplicated for an idea figure. It includes too many technical details, which makes it harder to quickly understand the main concept. A high-level illustration focusing only on the problem and the core idea would be clearer.

The main motivation sentence, “clinical descriptions are highly templated and semantically repetitive”, feels vague. I understand the general point, but it does not clearly explain why this is an actual problem for learning. If the issue is about false negatives in contrastive learning due to repetitive phrasing, it should be stated more explicitly.

Figure 2 is quite dense, and the overall pipeline is not very easy to follow. Some modules, like BCAM, appear in the diagram without a proper explanation. A short caption or visual cue describing what BCAM actually does and how it connects to other parts would help a lot.

The bidirectional fusion idea does not feel very new. Similar techniques have already been explored in earlier works such as [1] and [2]. In fact, the spatial relationship preserving approach here, where visual features are flattened into patch tokens, fused with text through cross-attention, and then reshaped back to the 2D grid, is essentially the same process described in [2]. The authors should clarify what is actually new in BCAM beyond this, for example, whether the “complementary role” assignment or the gating introduces any measurable improvement. A more detailed comparison with these existing techniques would help solidify the claimed novelty and make the contribution clearer.

Never clearly explain what defines a cluster or how those soft labels are derived or updated during training. It seems the text–text similarity matrix is computed once from a frozen encoder, which makes the “cluster-aware” term a bit misleading; here is no actual dynamic clustering or adaptive grouping during training.

The motivation for using KAN is not convincing. The paper does not explain why a spline-based gating function is needed or what specific limitation of existing nonlinearities like ReLU, GELU, or MLP-based gates it solves. As it stands, KAN feels like it was just added as-is without a strong reason. The improvement shown in the ablation study is also quite small, so it is hard to tell if the gain comes from KAN itself or simply from adding another nonlinear component. A clearer justification and a fair comparison with simpler gates would make this part more credible.

Since the entire claim revolves around better text–mask alignment, the paper should ideally visualize how text tokens influence spatial regions.

No formalized loss functions are described, nor is it explained how different objectives are balanced during training.

[1] Cho, Yubin, Hyunwoo Yu, and Suk-Ju Kang. "Cross-aware early fusion with stage-divided vision and language transformer encoders for referring image segmentation." IEEE Transactions on Multimedia 26 (2023): 5823-5833.

[2] Sultan, Rafi Ibn, et al. "BiPVL-Seg: Bidirectional Progressive Vision-Language Fusion with Global-Local Alignment for Medical Image Segmentation." arXiv preprint arXiv:2503.23534 (2025).

**Questions:**

Please address the points raised in the Limitations section and provide clarifications or responses to the concerns I posed there.

---

> ### Author Response · Authors · 2025-11-22
> **Response to Reviewer RBwK (Part 1/2)**
>
> Thank you very much for your thorough review and positive assessment of our work, and for the constructive comments that clearly highlight several shortcomings in the current manuscript. We respond to your points one by one below.
>
> 1. On the figures
>
> We agree with your comments regarding the figures. In the revised manuscript, we will redraw the figures to better emphasize the core motivation and the main pipeline. In subsequent versions, we will further refine their layout and readability, while keeping all essential information.
>
> 2. On making the motivation more concrete
>
> In Sec. 1 of the revised manuscript, we will add the following clarifications:
>
> (1) Quantifying the “templated / highly reused” phenomenon.
>
> For example, in QaTa-COV19, roughly 7,000 cases share only about 300 unique text descriptions. In other words, heavy text reuse within the same mini-batch is the norm rather than an occasional event during training. We will present this statistic explicitly in the introduction.
>
> (2) Explicitly linking “templating → false negatives → contrastive conflicts”.
>
> In the motivation section (together with Fig. 1), we will include a simple illustrative example showing how standard InfoNCE-style contrastive learning in such a setting naturally generates a large number of false negatives, which is harmful to the gradient-based learning of image–text alignment.
>
> (3) Clarifying that CaCL is specifically designed to alleviate these false negatives.
>
> CaCL uses a frozen text encoder to pre-compute an intra-batch text–text similarity matrix in the language space and applies row-mean debiasing and truncation to construct soft label distributions. As a result, the contrastive loss on image–text similarities no longer relies on “a single positive versus all remaining negatives”, but instead explicitly fits a continuous semantic similarity distribution. This directly mitigates the large number of false negatives caused by templated text reuse.
>
> 3. On the differences between BCAM and existing bidirectional fusion methods
>
> In many existing bidirectional interaction schemes, the so-called “language-dominant” branch only outputs updated text tokens, while the final segmentation prediction still relies on the visual feature maps. Text influences vision mainly through attention weights, without forming an explicit, language-centered spatial representation. This limits the ability to model fine-grained, text-conditioned spatial details.
>
> BCAM is designed to explicitly build complementary vision-dominant and language-dominant paths. In the language-dominant path, we maintain an explicit one-to-one relationship between tokens and spatial positions and avoid any fully connected compression or global pooling over the spatial dimension. This yields a language-dominant feature map that preserves spatial structure: each spatial location obtains a position-specific weighted combination over all text tokens, instead of being collapsed into a single global vector. This design helps retain small lesions and boundary details.
>
> In Fig. 5 of the revised manuscript, we add a visual comparison between a symmetric dual-path attention design (DualA) and BCAM under the same backbone and decoder. The results show that DualA’s attention responses are more prone to mismatch the textual description, whereas BCAM produces more concentrated responses around lesion boundaries and multiple lesion sites, consistent with its design goals. We believe this visualization will make BCAM’s advantage in spatial preservation more intuitive.
>
> At the same time, in Sec. 2 (Related Work) we will explicitly discuss the two representative papers you pointed out and clearly state that C2Seg builds on their insights, further emphasizing an explicit language-dominant spatial path plus a complementary role between the two branches.

---

> ### Author Response · Authors · 2025-11-22
> **Response to Reviewer RBwK (Part 2/2)**
>
> 4. On the term “cluster-aware”
>
> You rightly pointed out that our wording may give the impression that explicit clustering is performed during training, whereas in reality the similarity matrix is computed once in a frozen encoder space. This makes the term “cluster-aware” potentially misleading.
>
> What we mean by “cluster-aware” is that the loss function explicitly leverages local cluster structure in the language space to mitigate the false negatives induced by hard positive/negative sampling, not that we run an explicit clustering algorithm during optimization. In the revision, we will clarify this by rephrasing it as “leveraging implicit, batch-wise semantic neighborhoods in the frozen language space”, and adjust surrounding wording to avoid any suggestion of explicit clustering. Thank you for helping us recognize this potential ambiguity.
>
> 5. On KAN / K-Gate
>
> We agree with your concern that the original motivation and claimed benefits of K-Gate were somewhat overstated. In the revised manuscript, we will moderate the wording and reposition K-Gate as a lightweight, optional nonlinear gating module for fine-grained feature selection prior to modality fusion, rather than as a core innovation.
>
> The additional ablation in Table 4(a) shows that replacing a standard MLP gate with a KAN-based gate yields small but consistent performance gains with almost no extra parameters. We therefore consider K-Gate a meaningful low-cost enhancement, but will present it as such rather than as a central contribution.
>
> 6. On visualizing how text tokens influence spatial regions
>
> In the revised manuscript, we include activation visualizations for both DualA and BCAM (Fig. 5). As the figure shows, even when the prompts explicitly contain quantitative and positional cues such as “Unilateral”, “two”, and “lower left”, DualA still tends to produce results that contradict the text—for example activating both lungs or merging multiple lesions into a single region. In contrast, BCAM concentrates high-response regions more tightly within the text-specified target areas. This suggests that the language-dominant path effectively strengthens the mapping from textual semantics to spatial representations and, when combined with the vision-dominant path, not only alleviates text–mask inconsistency but also yields more accurate lesion localization and boundary delineation.
>
> For the original segmentation visualizations (Fig. 4), we will explicitly list the prompt text for each example (e.g., “unilateral pulmonary infection, one infected area”) and point out in which semantic dimensions (number, laterality, location) existing methods violate the text and how C2Seg avoids these mistakes. These failure and improvement cases, closely tied to textual semantics, will serve as intuitive evidence of improved text–mask consistency, complementing the gains observed in Dice/mIoU.
>
> We will incorporate these analyses into Sec. 4.2 and Sec. 4.3 to better illustrate how text tokens shape spatial regions and how our method achieves a deeper semantic understanding of the text to guide segmentation.
>
> 7. On the loss formulation and multi-objective weighting
>
> We apologize for not explicitly presenting the full loss function and weighting scheme in the original manuscript, which indeed makes it harder for readers to fully understand the training objective. In the revision, we will add the complete formulas and weights, roughly as follows:
>
> (1) In Stage I, we use a debiased text–text similarity matrix to construct soft labels and supervise the image→text and text→image similarity distributions with a contrastive loss, as given in Eq. (2).
>
> (2) In Stage II, we adopt a weighted combination of Dice loss and BCE loss. Dice loss directly optimizes segmentation quality from the perspective of region overlap and is particularly suitable in medical scenarios with small targets and class imbalance; BCE focuses on per-pixel classification correctness and provides smoother, more stable gradients. Their combination balances “pixel-level classification” and “region-level overlap” and is also a commonly used empirical choice in medical image segmentation.
>
> Once again, we sincerely thank you for your careful and insightful review. We have revised and supplemented the manuscript based on your feedback as described above. We believe these changes will help more accurately convey the actual contributions of this work and address your main concerns regarding clarity and originality.

---

> ### Comment · Reviewer_RBwK · 2025-11-27
>
> Thank you for the authors’ rebuttal; it clarified some of my concerns. However, most of these clarifications address oversights that should have been present in the original manuscript. Their response also overstates the novelty of BCAM by implying that prior works such as [1][2] do not perform bidirectional interaction in both paths, even though these methods already strengthen and complement both the visual and textual branches in ways similar to what the authors describe. While the authors addressed the presentation issues, many of their fixes rely heavily on promised revisions, and several of those revisions ultimately soften the original claims and further reduce the perceived novelty of the work. Hence, I will keep my original score: 4: marginally below the acceptance threshold. But would not mind if paper is accepted.
>
> [1] Cho, Yubin, Hyunwoo Yu, and Suk-Ju Kang. "Cross-aware early fusion with stage-divided vision and language transformer encoders for referring image segmentation." IEEE Transactions on Multimedia 26 (2023): 5823-5833.
>
> [2] Sultan, Rafi Ibn, et al. "BiPVL-Seg: Bidirectional Progressive Vision-Language Fusion with Global-Local Alignment for Medical Image Segmentation." arXiv preprint arXiv:2503.23534 (2025).

---

### Official Review · Reviewer_RDjv · 2025-10-30

**Soundness:** 2
**Presentation:** 3
**Contribution:** 2
**Rating:** 2
**Confidence:** 4

**Summary:**

This paper proposes C2Seg, a two-stage vision–language framework for medical image segmentation, aiming to improve alignment between textual descriptions and segmentation masks. The first stage introduces Cluster-aware Contrastive Learning (CaCL), which converts intra-batch text similarities into soft labels for contrastive supervision. The second stage adds a Bidirectional Complementary Attention Module (BCAM) that integrates vision- and language-dominant paths, followed by a KAN-based Attention Gating (K-Gate) for nonlinear feature selection.

**Strengths:**

1. The paper is well written, logically structured, and easy to follow.
2. The topic of improving text–mask alignment in multimodal segmentation is relevant to medical AI.
3. The authors conduct detailed ablation studies and provide reproducibility information.

**Weaknesses:**

1. The methodological novelty is limited, and the contribution appears primarily engineering-oriented. CaCL functions as a soft-label variant of conventional contrastive learning, closely aligned with supervised or semantic-aware contrastive paradigms. BCAM represents only a modest architectural modification to existing bidirectional cross-attention designs, while K-Gate simply replaces standard MLPs with KAN units without clear theoretical motivation or demonstrated substantive benefit. Overall, the paper assembles several established components—soft contrastive learning, dual-path attention, and nonlinear gating—without offering a cohesive conceptual advance or deeper theoretical insight.
2. The central problem of “text–mask consistency” is not clearly defined. The paper lacks a formal metric or quantitative measure beyond Dice and mIoU, making it unclear whether the improvement truly reflects better semantic alignment or merely better segmentation accuracy.
3. The experimental scope is narrow, relying solely on two COVID-related datasets with templated or synthetic text, and both tasks are limited to binary lesion segmentation. This setup is not representative of real clinical scenarios and does not evaluate the method under more challenging multi-class or multi-organ settings. As a result, the proposed approach remains highly restricted in scope, and the experiments do not demonstrate generalization to other organs, modalities, or real free-text clinical reports.
4. The reported performance improvements are modest (≈1–2% Dice) compared with the additional complexity introduced by multi-stage training and multiple attention and gating modules. The computational cost–benefit trade-off is not justified.

**Questions:**

See weaknesses.

---

> ### Author Response · Authors · 2025-11-22
> **Response to Reviewer RDjv (Part 1/2)**
>
> Thank you very much for your careful review of our work. We address your concerns point by point below.
>
> 1. On the novelty of the method
>
> We agree with your observation that CaCL, BCAM, and K-Gate are built upon established ideas such as contrastive learning, bidirectional attention, and nonlinear gating. Our goal is precisely to clearly position this work as a task-driven, problem-specific design and systematic empirical study around text–mask inconsistency in medical vision–language segmentation, rather than a reinvention of contrastive learning or attention itself.
>
> More specifically:
>
> (1) Role of CaCL
>
> We do not claim that CaCL is fundamentally new at the level of contrastive learning paradigms. Rather, we aim to show how a mature contrastive learning mechanism can be adapted to the specific setting of highly templated and heavily reused medical text. Mainstream contrastive methods typically adopt one-to-one matching; even when extended to “one-to-many”, they still rely on hard positive/negative labels, making it difficult to explicitly model a continuous semantic similarity spectrum among samples.
>
> In contrast, CaCL constructs an intra-batch text–text similarity matrix in a frozen text-encoder space to characterize the “semantic neighborhoods” among templated reports, and uses this to generate soft labels. This allows us to explicitly handle the case where the same textual description corresponds to multiple different images, so that these pairs are no longer all treated as strong negatives.
>
> (2) Role of BCAM
>
> The starting point of BCAM is to explicitly construct complementary vision-dominant and language-dominant paths, while preserving fine-grained spatial structure in the language path.
>
> In many existing “bidirectional interaction” designs, the so-called language-dominant branch only outputs updated text tokens, while the final segmentation prediction still relies on the visual feature maps. Text mainly influences vision indirectly via attention weights, rather than forming an explicit, language-centered spatial representation. As a result, the model’s ability to capture fine-grained, text-conditioned semantics such as lesion count, laterality, and coarse location is limited.
>
> In BCAM’s language-dominant path, we maintain an explicit correspondence between each token and spatial positions, avoiding fully connected compression or global pooling along the spatial dimension. Each spatial position receives a position-specific weighted combination over all text tokens rather than being collapsed into a single global vector. This directly outputs a spatial feature map that is language-dominant yet structure-preserving, which helps retain small lesions and boundary details and complements the vision-dominant path.
>
> (3) Role of K-Gate
>
> K-Gate is our mechanism for modality-wise feature selection and is intended as a helpful but optional nonlinear gate. As shown in Table 4, using KAN for gating yields small but consistent improvements over a standard MLP gate, with negligible additional parameters. In the revision we will tone down the wording and emphasize that the value of this work lies in designing and validating a task-specific solution around a realistic but under-explored problem—text–mask inconsistency—rather than in K-Gate itself as a standalone theoretical contribution.
>
> 2. On the definition and measurement of “text–mask consistency”
>
> We appreciate your comment that the definition of “text–mask consistency” in the original manuscript was not sufficiently explicit. In our setting, text–mask consistency is not meant as an abstract notion; it refers to whether the predicted mask agrees with the prompt text along several key semantic attributes, such as:number of lesions, laterality (left lung / right lung / bilateral), coarse spatial location (upper / middle / lower zone, or specific lobes), and so on.
>
> For example:
>
> (1) If the text states “unilateral pulmonary infection, one infected area”, but the predicted mask covers both lungs or contains multiple disconnected foci, then even if the Dice score is not particularly low, we would consider this a text–mask inconsistency;
>
> (2) If the text states “two infected areas”, but the prediction merges them into one large contiguous region, then the prediction contradicts the text in terms of lesion count, even if overall overlap is good.
>
> We will add this explicit, attribute-level definition of text–mask consistency in Sec. 1 of the revised manuscript.

---

> ### Author Response · Authors · 2025-11-22
> **Response to Reviewer RDjv (Part 2/2)**
>
> We fully agree that Dice/mIoU alone cannot faithfully measure such consistency. During the rebuttal we did experiment with simple proxy metrics based on connected-component counts and heuristic left/right lung partition to estimate agreement between “text-described number/laterality” and the predicted mask. However, due to the lack of fine-grained structured labels in current public datasets (e.g., exact lesion counts, lobe-level annotations), these proxy metrics turned out to be highly sensitive to hand-crafted rules and thresholds and not sufficiently stable across datasets.
>
> For reasons of rigor, we decided not to introduce a new quantitative metric that has not been thoroughly validated, so as not to give the impression that such an index is already mature and reliable. Instead, we treat a systematic quantitative metric for text–mask consistency as an important direction for future work. In the current version, we will strengthen the qualitative evidence by:
>
> (1) In Sec. 4.2, explicitly writing out the prompt text for each visualization (e.g., “unilateral left lung with one infected area”, “two lesions in the middle and lower zones of the left lung”), and pointing out precisely where existing methods fail in terms of number, laterality and location, and how C2Seg avoids these errors under the same prompts;
>
> (2) Using these failure/improvement cases—strongly tied to textual semantics—as intuitive evidence of text–mask consistency, in conjunction with the improvements in Dice/mIoU/HD95/ASSD.
>
> We will also clearly state in the conclusion and outlook that a systematic, general-purpose quantitative metric for text–mask consistency remains an open problem requiring richer annotations and task settings, and that this is one of the main directions of our future work.
>
> 3. On the experimental scope
>
> We have added two public gastrointestinal endoscopic polyp segmentation datasets, CVC-ClinicDB and Kvasir. Both target GI polyp segmentation and differ substantially from chest CT in imaging modality, organ structure and lesion appearance. Image–text pairs for these datasets follow the clinically styled descriptions provided by Zhang et al.~[1].
>
> As shown in Table 2 of the revised manuscript, C2Seg remains clearly superior to multiple representative baselines on both datasets, indicating that our design is not limited to lung CT but exhibits a certain degree of generalization across organs and imaging modalities.
>
> In addition, beyond Dice/mIoU, we now report HD95 (95th percentile Hausdorff Distance) and ASSD (Average Symmetric Surface Distance) to more comprehensively evaluate contour position and boundary detail. The results show that our method also provides consistent gains on these distance-based metrics, suggesting that in addition to improving overall overlap, C2Seg yields more accurate lesion boundaries and spatial localization.
>
> 4. On the magnitude of performance gains and complexity–benefit trade-off
>
> CaCL operates in Stage I as a contrastive pretraining objective and does not alter the network architecture or introduce additional parameters; it can be removed during inference. BCAM is a dual-path module, but the two paths share one set of cross-attention weights, and at each scale we perform only one image→text and one text→image cross-attention, unlike some approaches that stack many layers of self-attention plus repeated cross-attention. K-Gate is a narrow KAN-based gate; compared to a standard MLP, the additional parameters and computation are very modest.
>
> Thus, the extra overhead of C2Seg at deployment/inference time is relatively small. To address your concern, we have added the parameter counts of all methods to Tables 1 and 2 in the revised manuscript. Our C2Seg model has about 18.92M parameters, which is on the lighter side among the compared methods. We believe that achieving around 1–2% Dice improvement on strong baselines while remaining relatively lightweight is practically meaningful, especially given the difficulty of further gains in this regime.
>
> We have revised and supplemented the manuscript according to the above points. We hope these changes present the actual contribution of our work more accurately and help alleviate your concerns.
>
> Reference
>
> [1] Zhang X, Ni B, Yang Y, et al. Madapter: A better interaction between image and language for medical image segmentation. In MICCAI, 2024.

---

> > ### Comment · Reviewer_RDjv · 2025-11-27
> >
> > Thank you for the authors’ rebuttal. The response has clarified several implementation details and has partially addressed some of my earlier questions. However, my primary concerns remain unresolved.
> >
> > First, regarding methodological novelty, in the rebuttal, the authors reiterate that these components are task-driven adaptations rather than fundamentally new mechanisms. Still, this clarification does not address my core concern.
> >
> > Second, regarding experimental scope, although two additional datasets were included, they are still relatively simple binary lesion segmentation tasks and do not further demonstrate the method’s potential applicability to broader and more complex real clinical scenarios, e.g., multi-class settings.
> >
> > Given these unresolved concerns, I will maintain my original score: Rating: 2 — reject, not good enough.

---

### Official Review · Reviewer_bmue · 2025-10-30

**Soundness:** 2
**Presentation:** 3
**Contribution:** 2
**Rating:** 4
**Confidence:** 4

**Summary:**

This paper tackles text-mask inconsistency in medical image segmentation through C2Seg, featuring Cluster-aware Contrastive Learning (CaCL) and Bidirectional Complementary Attention Module (BCAM). The motivation is clear and results show improvements, but I have concerns about novelty and experimental scope.

**Strengths:**

- The problem is well-motivated—templated clinical text causing false negatives in contrastive learning is genuine, nicely illustrated in Figure 1.

- Soft-label construction using text-text similarity is intuitive with clear mathematical formulation in Equations 1-3.

- BCAM's spatial structure preservation addresses M3Att's limitation where fully connected projection loses local details.

**Weaknesses:**

The "frozen strong baseline" for computing text-text similarity lacks justification—medical text has domain-specific semantics that general CLIP might not capture well [1], yet there's no sensitivity analysis to different encoders like BioBERT or PubMedBERT. The row-mean debiasing M'ij = max{Mij - μi, 0} also needs theoretical grounding since hard thresholding might remove valid semantic structure [2], and the gradient analysis in Equation 3 is purely conceptual without quantitative validation of how much false negative attenuation actually occurs. The language-dominant path in Equation 6 sums over N tokens without normalization which should cause unstable magnitudes across varying text lengths, and BCAM's spatial preservation claim lacks attention visualizations to verify spatially-coherent patterns compared to M3Att.

Novelty is limited since soft-label contrastive learning already exists in Prototypical Contrastive Learning [3], ProtoNCE [4], and medical-specific MedKLIP [5]. BCAM is essentially incremental modification of M3Att with changed spatial handling, and KAN integration is straightforward MLP substitution without domain-specific adaptation—Table 3 even shows pure KAN encoder performs worse than hybrid, contradicting claims about superior nonlinear modeling. The overall framework combines existing techniques without fundamentally new mechanisms for text-mask consistency.

Evaluation is severely limited to two COVID chest imaging datasets, preventing generalization claims to diverse anatomies like brain MRI, abdominal CT, or pathology [6]. Missing comparisons with recent medical VLMs like LLaVA-Med, BiomedCLIP, SAM-Med, and MedSAM that directly target medical segmentation. No analysis of text quality robustness when reports have typos, abbreviations, or incomplete descriptions [7], and no computational cost reporting despite CaCL adding O(B²C) complexity and BCAM doubling attention paths—training time, memory, inference speed are all absent. Hyperparameter choices (ρ=0.8, τ=0.07, batch size 256) appear arbitrary without sensitivity analysis, and the "text reuse" problem (7000 samples sharing 300 descriptions) is mentioned but never correlated with performance gains.


### Referenced Works

[1] Müller et al., "Joint Learning of Localized Representations from Medical Images and Reports", Medical Image Analysis, 2022

[2] Khosla et al., "Supervised Contrastive Learning", NeurIPS, 2020

[3] Li et al., "Prototypical Contrastive Learning of Unsupervised Representations", CVPR, 2021

[4] Dwibedi et al., "With a Little Help from My Friends: Nearest-Neighbor Contrastive Learning of Visual Representations", ICCV, 2021

[5] Wu et al., "Medical Knowledge-enhanced Large Language Model", Nature Medicine, 2024

[6] Ma et al., "Segment Anything in Medical Images", MICCAI, 2024

[7] Bozkurt et al., "Reporting Standards for Clinical Decision Support", Nature Digital Medicine, 2023

**Questions:**

Can you provide empirical evidence for CaCL's effectiveness by measuring actual false negative rates, gradient magnitudes, or negative pair distances before/after, and show sensitivity analysis for frozen encoder choice across domain-specific alternatives? The current analysis is purely conceptual without quantitative validation.

Why does Table 3 show pure KAN encoder underperforming hybrid design, and what's the rationale for hyperparameters (ρ=0.8, τ=0.07, batch size 256) without sensitivity analysis? How does text length variation affect the unnormalized summation in Equation 6?

Where are attention visualizations verifying BCAM's spatial preservation, failure case analyses for multi-site/multi-lesion scenarios, and computational cost comparisons (training time, memory, FLOPs) for practical deployment? How does performance degrade with noisy or incomplete text descriptions common in real clinical settings?

---

> ### Author Response · Authors · 2025-11-22
> **Response to Reviewer bmue (Part 1/3)**
>
> Thank you very much for your careful review and constructive comments. We respond to the main points as follows.
>
> 1. On CaCL
>
> (1) Why we adopt a generic CLIP-style encoder, and sensitivity analysis
>
> We choose a CLIP-style large-scale multimodal text encoder as the frozen backbone because the textual descriptions in our setting are short, templated, and relatively coarse-grained (e.g., “bilateral pulmonary infection”, “three infected areas”). Prior work has shown that general-purpose multimodal models are reliable for semantic extraction on such simple sentences.
>
> To address your concern about domain-specific encoders, we conducted an additional sensitivity study during the rebuttal phase: keeping the CaCL framework and training strategy unchanged, we encoded text with CLIP / BioBERT / PubMedBERT respectively and retrained on QaTa-COV19 and MosMedData+. The results show that the performance differences are small and CLIP is slightly better. We will include this comparison table in Sec. 4.3, and emphasize that C2Seg is not overly sensitive to the specific text encoder, and using CLIP as the default choice is a reasonable and slightly favorable engineering decision.
>
> (2) Motivation of row-mean debiasing and clipping
>
> Under highly templated clinical descriptions, if we directly use the raw cosine similarity matrix M, many entries in a given row tend to be uniformly high due to shared templates, leading to the phenomenon that “all negatives look similar” and making the soft labels poorly reflect the true semantic structure. Therefore, CaCL applies row-wise mean debiasing and non-negative clipping to each row.
>
> Subtracting the row mean (template bias) reduces the global similarity inflation introduced by shared templates and highlights semantic differences on top of that template. The subsequent non-negative clipping encourages the soft labels to focus on local semantic neighborhoods rather than spreading weight almost uniformly across all samples. We will add this explanation in Sec. 3.2 of the revised manuscript.
>
> (3) Empirical effect of CaCL
>
> In the original ablation, we already show that removing CaCL or replacing CaCL with vanilla contrastive learning consistently degrades segmentation performance on both datasets, which serves as direct empirical evidence of its effectiveness. We will make this point more explicit in the revision. At the same time, we will add a discussion comparing CaCL with Prototypical CL / ProtoNCE and related approaches in the related work section.
>
> Regarding the statement that “about 7,000 samples share 300 descriptions”, our intention was to highlight that under such a text distribution, the same sentence is very likely to be repeatedly sampled as a negative within a batch, causing standard InfoNCE to systematically generate a large number of false negatives and distort gradients. CaCL uses soft labels to assign intermediate weights to those “same text but different image” pairs, thereby alleviating this conflict at the methodological level.
>
> 2. On hyperparameter settings and normalization in Eq.(6)
>
> (1) Sensitivity of ρ, τ, and batch size
>
> We agree that hyperparameter choices should be supported by sensitivity analysis. During the rebuttal, we performed systematic experiments on MosMedData+ with:
>
> · ρ∈{0.6,0.8,1.0};
>
> · τ∈{0.05,0.07,0.10};
>
> · batch size∈{128,256,512}.
>
> The results show that ρ=0.8 and τ=0.07 yield the best performance; decreasing or increasing these values leads to slightly worse results, but the differences remain modest, indicating that the model is not overly sensitive to these hyperparameters. For batch size, larger batches indeed help to estimate intra-batch semantic structure more stably, but performance already saturates around 256. We will include these sensitivity results in Table 4 of the revised manuscript and clarify that the final settings are a compromise between performance and GPU memory constraints.

---

> ### Author Response · Authors · 2025-11-22
> **Response to Reviewer bmue (Part 2/3)**
>
> (2) Unnormalized summation in Eq.(6) and text length
>
> You correctly point out that directly summing over N tokens in Eq.(6) could cause magnitude variations with changing text length. In our current datasets, however, the clinical descriptions are highly templated and, after preprocessing (truncation/padding), the actual token lengths are almost uniform. Together with subsequent LayerNorm and other normalization layers, we did not observe noticeable numerical instability in the original implementation.
>
> That said, we fully agree that adding explicit normalization at the formula level is more rigorous. In the revised manuscript, we have rewritten Eq.(6) to explicitly normalize over the token dimension and updated the code accordingly. The new implementation yields almost identical Dice and mIoU on QaTa-COV19 and MosMedData+ (both changes < 0.1%). In future work, we plan to further study the effect of free-form clinical text with more variable length, so as to extend our approach to richer clinical reporting scenarios.
>
> 3. On individual components
>
> (1) CaCL
>
> We will not present CaCL as a “theoretical breakthrough in contrastive learning”, but explicitly position it as a targeted adaptation of existing contrastive learning methods to the setting of highly templated and heavily reused medical text. Traditional contrastive learning largely relies on hard one-to-one labels; even when extended to “one-to-many”, the annotation scheme still imposes a hard positive/negative split, making it difficult to model a continuous spectrum of semantic similarities among samples.
>
> CaCL leverages an intra-batch text–text similarity matrix in the frozen language space to characterize the “semantic neighborhood” among templated reports, and uses this to explicitly handle the case where the same text is paired with multiple different images.
>
> (2) BCAM
>
> The design goal of BCAM is to explicitly build complementary vision-dominant and language-dominant paths. In the language-dominant path, we always maintain a one-to-one correspondence between each token and a spatial position, and avoid fully connected compression or global pooling along the spatial dimension. As a result, the language branch directly outputs a spatial feature map that preserves the image structure, where each spatial position receives a token-specific weighted combination over all text tokens rather than being collapsed into a single global vector. This helps retain differences in small lesions and boundaries.
>
> In Fig. 5 of the revised manuscript, we visualize and compare attention maps produced by the symmetric DualA design and BCAM under the same backbone and decoder. The results show that DualA tends to produce responses that are more often misaligned with the textual description, whereas BCAM yields more concentrated activations around lesion boundaries and multi-focal lesions, which is consistent with its design goal. We believe this visualization provides an intuitive demonstration of the spatial-preserving advantage of BCAM.
>
> (3) KAN and K-Gate
>
> We appreciate your pointing out that our original wording around KAN/K-Gate was somewhat too strong. K-Gate is intended as a helpful but optional nonlinear gating mechanism for modality-wise feature selection. The ablations in Table 4 show that using KAN for gating yields small but consistent gains over a standard MLP gate, with negligible parameter overhead. We will tone down the original claims and weaken statements about “interpretability/universality”.
>
> Regarding your concern that a pure KAN encoder underperforms the hybrid design, which seems at odds with “stronger nonlinear modeling”, we clarify this as follows: as discussed in Sec. 4.3, CNNs provide critical local receptive fields for low-level feature extraction, and local texture and structural priors are particularly important in medical imaging. The results in Table 3 precisely show that “CNN + KAN hybrid > CNN only or KAN only”, which is why we adopt a hybrid design in both the encoder and gating. In the revision, we will explicitly state that KAN is better suited as a complement to CNNs rather than a complete replacement, and rephrase overly absolute expressions (e.g., “superior nonlinear modeling”) into more neutral ones such as “provides complementary nonlinear modeling capacity with modest gains”.

---

> ### Author Response · Authors · 2025-11-22
> **Response to Reviewer bmue (Part 3/3)**
>
> 4. On experimental scope, computational cost, and robustness to text quality
>
> (1) Experimental scope
>
> We have added two public datasets with very different imaging modalities and anatomy: CVC-ClinicDB and Kvasir, both corresponding to gastrointestinal endoscopic polyp segmentation. Image–text pairs are constructed following Zhang et al.~[1], whose descriptions are clinically oriented and written in a style similar to radiology/endoscopy reports.
>
> As shown in Table 2 of the revised manuscript, C2Seg consistently outperforms multiple representative baselines on these two datasets as well, suggesting that our design is not restricted to lung CT, but exhibits a certain degree of generalization across organs and modalities. We will also explicitly acknowledge in the Limitations that the current experiments mainly involve short textual prompts, and that extending to brain MRI, abdominal CT, pathology images, and richer multi-label reports is an important direction for future work.
>
> (2) Computational cost
>
> The main additional cost of C2Seg comes from the attention-based fusion. Compared with some heavy architectures that stack multiple layers of self-attention and repeated cross-attention, our bidirectional module runs only one image→text and one text→image cross-attention per scale, with the two paths sharing the same attention matrices. The overall model has about 18.92M parameters, which is on the lower end among compared methods. We list the parameter counts of all baselines in Tables 1 and 2 of the revised manuscript.
>
> (3) Relation to medical VLMs and robustness to text noise
>
> Regarding recent medical VLMs such as LLaVA-Med, BiomedCLIP, SAM-Med, and MedSAM, we will expand the discussion in Sec. 2 of the related work. In general, these models focus more on providing powerful multimodal semantic representations and large-model interfaces, while C2Seg is positioned as a task-level training and fusion framework that can be plugged into different VLM encoders. In this work, we focus on addressing text–mask inconsistency given a chosen encoder, rather than directly competing with those large models.
>
> We also acknowledge that the textual descriptions in our datasets are relatively clean and well structured, which is not sufficient to fully assess robustness to noisy clinical reports. In the Limitations section, we will explicitly point out that systematically studying C2Seg under realistic text noise—such as spelling errors, abbreviations, and missing findings—is an important but non-trivial direction beyond the scope of this paper, and we plan to investigate it in future work using real hospital data and richer text augmentation strategies.
>
> We have revised and supplemented the manuscript according to the above points. We believe these changes help present the actual contributions of this work more accurately and hope they address your main concerns.
>
> Reference
>
> [1] Zhang X, Ni B, Yang Y, et al. Madapter: A better interaction between image and language for medical image segmentation. MICCAI, 2024.

---

### Official Review · Reviewer_WFDv · 2025-11-01

**Soundness:** 3
**Presentation:** 3
**Contribution:** 3
**Rating:** 6
**Confidence:** 4

**Summary:**

This paper proposed a text-mask consistency enhanced two stage segmentation framework (with CaCL, BCAM, and K-Gate). The experiments have been done on two COVID datasets and showed clear improvement.

**Strengths:**

The methodology is sound and the paper is well-structured.
The paper’s motivation is clearly articulated by identifying specific causes of text–mask inconsistency, which justifies the two-stage approach.
The results show clear improvements in Dice and mIoU, and the ablation studies are thorough, which lends credence to the methodological claims.

**Weaknesses:**

1) The proposed C2Seg framework (with CaCL, BCAM, and K-Gate) appears to be a thoughtful integration.  While each component has roots in existing techniques (e.g. contrastive learning and dual attention are established concepts, and KANs have been explored in vision backbones), which require more detailed clarification on the technical contribution.
2) The experiments use two public medical imaging datasets (QaTa-COV19 and MosMedData+), both are in the COVID-19 chest imaging domain, so the representativeness is somewhat narrow.
3) One minor critique is that the evaluation emphasizes standard segmentation metrics without a direct quantitative measure of “text–mask consistency”, but qualitatively the improvements imply better alignment.

**Questions:**

please address the comments in weakness session. clarify the technical novelty, add more datasets, and maybe more evaluation metrics.

---

> ### Author Response · Authors · 2025-11-22
> **Response to Reviewer WFDv (Part 1/2)**
>
> Thank you very much for your positive comments on our work and for the valuable suggestions. We respond to your points one by one as follows.
>
> 1. Clarifying the technical contributions (CaCL, BCAM, K-Gate)
>
> (1) CaCL.
>
> We do not claim that CaCL itself is a fundamentally new contrastive learning paradigm; instead, we position it as a task-driven adaptation for medical scenarios. Under highly templated and heavily reused clinical descriptions, mainstream contrastive learning typically adopts one-to-one matching. Even when extended to “one-to-many”, these methods still rely on hard positive/negative labels at the annotation level, making it difficult to explicitly capture a continuous spectrum of semantic similarities between samples.
>
> The core idea of CaCL is to construct an intra-batch text–text similarity matrix in the frozen language encoder space to characterize the “semantic neighborhood” among templated reports, and to use this structure to generate soft labels. In this way, when the same text appears with multiple different images, these pairs are no longer all treated as strong negatives. In other words, CaCL does not invent a new contrastive loss from scratch; rather, it provides a contrastive supervision mechanism that is better aligned with the semantic geometry induced by highly templated, heavily reused medical text—an issue that has received relatively little systematic attention.
>
> (2) BCAM.
>
> The starting point of BCAM is to explicitly construct complementary vision-dominant and language-dominant paths, while preserving as much fine-grained spatial structure as possible.
> In many existing “bidirectional interaction” designs, the so-called “language-dominant” branch typically outputs only updated text tokens, while the final segmentation prediction still relies on the visual branch features. In this regime, text mainly influences visual features indirectly through attention weights, without forming an explicit, language-centered spatial representation, which limits the modeling of fine-grained semantics such as lesion count, laterality, and coarse spatial location.
>
> In the language-dominant path of BCAM, we always maintain an explicit correspondence between each token and a particular spatial position, and we no longer compress or globally pool the spatial dimension. Each spatial position receives a position-specific weighted combination over all text tokens, instead of being collapsed into a single global vector. This design allows the language branch to produce a spatial feature map that preserves the underlying image structure, which is more favorable for modeling small lesions and boundary details, and complements the vision-dominant path.
>
> (3) K-Gate.
>
> We sincerely appreciate your comment that our original description of K-Gate was somewhat overstated. In the revision, we no longer present K-Gate as a major theoretical contribution. Instead, we more cautiously position it as a lightweight, optional nonlinear gating module used to suppress noise and highlight informative modalities before fusion. The additional ablation in Table 4(a) shows that, compared to a standard MLP gate, using KAN for gating yields stable but moderate performance gains with almost no additional parameters. In the revised manuscript, we will tone down related claims and present K-Gate as a “helpful but non-core” auxiliary component.
>
> Summary.
>
> In the revision (especially in Sec. 1 and at the beginning of the Method section), we will provide a more focused clarification of these three aspects, and highlight the central contribution of this work: we build on well-established contrastive learning and bidirectional attention mechanisms, but systematically adapt and redesign them around the problem of text–mask inconsistency in medical scenarios, rather than simply stacking existing techniques.

---

> ### Author Response · Authors · 2025-11-22
> **Response to Reviewer WFDv (Part 2/2)**
>
> 2. On dataset scope, generalization, and evaluation metrics
>
> We fully agree with your concern that using only two COVID chest datasets may limit the representativeness of our experiments. To address this, during the rebuttal period we added two public datasets from a different organ and imaging modality: CVC-ClinicDB and Kvasir:
>
> (1) Both datasets come from gastrointestinal endoscopic polyp segmentation, which differs substantially from chest CT in terms of imaging modality and anatomical structure;
>
> (2) Their image–text pairs are constructed following Zhang et al.~[1], where the textual descriptions are clinically meaningful and written for physician interpretation;
>
> (3) As shown in the newly added results in Table 2 of the revised manuscript, C2Seg still clearly outperforms multiple representative methods on these two polyp datasets. This suggests that our approach is not restricted to chest CT, but also exhibits a certain level of generalization across organs and imaging conditions.
>
> In addition, we have introduced two boundary/distance-related metrics—HD95 (95% Hausdorff Distance) and ASSD (Average Symmetric Surface Distance)—to more comprehensively evaluate the quality of the predicted masks in terms of contour location and geometric details. Experimental results show that C2Seg also achieves consistent improvements on HD95 and ASSD: not only does it obtain higher region-overlap scores (Dice / mIoU), but it also yields more accurate lesion boundaries and spatial localization. This further supports, from a geometric perspective, the effectiveness of using text constraints to guide spatial localization.
>
> 3. On quantitative measures of “text–mask consistency”
>
> We fully agree with your point that Dice and mIoU alone are not sufficient to fully capture “text–mask consistency”. During the rebuttal, we attempted to design simple proxy metrics based on lesion count, laterality (left/right lung), etc., but encountered the following issues:
>
> (1) Current public datasets lack comprehensive structured annotations (e.g., precise lesion counts, lobe-level labels);
>
> (2) Simple heuristics based on connected-component counting or manually defined left/right lung partitions are highly sensitive to thresholds and rules, and their stability across datasets is limited.
>
> Out of rigor, we decided not to introduce such preliminary metrics in the current version, so as not to give the impression that they are already well-validated and widely applicable. Instead, in the revised manuscript we will strengthen the analysis and presentation of text–mask consistency from two complementary angles:
>
> (1) In Sec. 4.2, we will explicitly analyze failure cases along three semantic dimensions—count, laterality, and coarse location—showing typical errors made by existing methods under the same textual prompts, and how C2Seg avoids these errors.
>
> (2) We will relate these text-sensitive visual examples to the improvements in Dice / mIoU / HD95 / ASSD, so that readers can more intuitively see how gains in “semantic consistency” are reflected in both geometric and overlap-based metrics.
>
> At the same time, we will explicitly state in the Limitations section that: a systematic and broadly applicable quantitative measure of text–mask consistency remains an open problem, and will require richer annotations and real clinical scenarios for dedicated study. This will be one of the main directions of our future work.
>
> We have already incorporated the above clarifications and additions into the revised manuscript. We believe these changes help present the actual contributions of this work more accurately, and we hope they can help to alleviate some of your concerns.
>
> Reference
>
> [1] Zhang X, Ni B, Yang Y, et al. Madapter: A better interaction between image and language for medical image segmentation. MICCAI, 2024.

---

### Meta-Review · Area_Chair_WFrH · 2026-01-03

**Summary:**

C2Seg seeks text–mask consistency in medical image segmentation through “soft-contrast pre-training” + “bidirectional complementary attention” + “KAN-based non-linear gating.” The motivation is clear and the results show improvements. However, the authors failed to address most of the reviewers’ concerns, so the rebuttal outcome was far from satisfactory.

**Reviewer Concerns:**

During the rebuttal, the authors elaborated on methodological and experimental details, evaluated C2Seg on two additional datasets, and clarified some of the reviewers’ questions. Nevertheless, they did not fully resolve the reviewers’ concerns regarding the novelty of the approach and the row-mean debiasing strategy.

**Reviewer Scores:**

The majority of reviewers explicitly stated that they would not raise their scores.

---

### Decision · Program_Chairs · 2026-01-26

Accept (Poster)